# Measurement report: Dual-carbon isotopic characterization of carbonaceous aerosol reveals different primary and secondary sources in Beijing and Xi'an during severe haze events

Haiyan Ni[1,2], Ru-Jin Huang[1,3], Max M. Cosijn[2], Lu Yang[1], Jie Guo[1], Junji Cao[1], and Ulrike Dusek[2]

[1]State Key Laboratory of Loess and Quaternary Geology, Center for Excellence in Quaternary Science and Global Change, Key Laboratory of Aerosol Chemistry and Physics, Institute of Earth Environment, Chinese Academy of Sciences, Xi'an, 710061, China
[2]Centre for Isotope Research (CIO), Energy and Sustainability Research Institute Groningen (ESRIG), University of Groningen, Groningen, 9747 AG, the Netherlands
[3]Institute of Global Environmental Change, Xi'an Jiaotong University, Xi'an, 710049, China

*Correspondence*: Ru-Jin Huang (rujin.huang@ieecas.cn)

**Abstract.** To mitigate haze pollution in China, a better understanding of the sources of carbonaceous aerosols is required due to the complexity in multiple emissions and atmospheric processes. Here we combined the analysis of radiocarbon and the stable isotope $^{13}C$ to investigate the sources and formation of carbonaceous aerosols collected in two Chinese megacities (Beijing and Xi'an) during severe haze events of "red alarm" level from December 2016 to January 2017. The haze periods with daily $PM_{2.5}$ concentrations as high as ~400 µg m$^{-3}$ were compared to subsequent clean periods (i.e., $PM_{2.5}$ < median concentrations during the winter 2016/2017), with $PM_{2.5}$ concentrations below 100 µg m$^{-3}$ in Xi'an and below 20 µg m$^{-3}$ in Beijing. In Xi'an, liquid fossil fuel combustion was the dominant source of elemental carbon (EC; 44%–57%), followed by biomass burning (25%–29%) and coal combustion (17%–29%). In Beijing, coal combustion contributed 45%–61% of EC and biomass burning (17%–24%) and liquid fossil fuel combustion (22%–33%) contributed less. Non-fossil sources contributed 51%–56% of organic carbon (OC) in Xi'an and fossil sources contributed 63%–69% of OC in Beijing. Secondary OC (SOC) was largely contributed by non-fossil sources in Xi'an (56 ± 6%) and by fossil sources in Beijing (75 ± 10%), especially during haze periods. The fossil vs. non-fossil contributions to OC and EC did not change drastically during haze events in both Xi'an and Beijing. However, compared to clean periods, the contribution of coal combustion to EC during haze periods increased in Xi'an and decreased in Beijing. During clean periods, primary OC from biomass burning and fossil sources constituted ~70% of OC in Xi'an and ~53% of OC in Beijing. From clean to haze periods, the contribution of SOC to total OC increased in Xi'an, but decreased in Beijing, suggesting that contribution of secondary organic aerosol formation to increased OC during haze periods was more efficient in Xi'an than in Beijing. In Beijing, the high SOC fraction in total OC during clean periods was mainly due to elevated contribution from non-fossil SOC. In Xi'an, a slight day-night difference was observed during the clean period, with enhanced fossil contributions to OC and EC during the day. This day-night difference was negligible during severe haze periods, likely due to enhanced accumulation of pollutants under stagnant weather conditions.

## 1 Introduction

Severe haze pollution with high $PM_{2.5}$ (i.e., particulate matter with aerodynamic diameter $\leq 2.5\,\mu m$) concentrations and reduced visibility occurs frequently during winter in China (An et al., 2019). Field measurements show that carbonaceous aerosol contributes a significant fraction of $PM_{2.5}$ loading during severe haze events in China (Huang et al., 2014; Elser et al., 2016; Liu et al., 2016). Therefore, a better understanding of the sources and atmospheric processes of carbonaceous aerosols is needed for mitigating haze pollution. Many previous studies focus solely on Beijing, the capital of China. However, studies on other megacities are also needed for comparison as well as for a more comprehensive understanding of haze pollution in China.

Carbonaceous aerosol constituents are separated into elemental carbon (EC) and organic carbon (OC), fractions differing in their thermal refractiveness with EC being thermally refractory and OC weakly refractory (Pöschl, 2003, 2005; Petzold et al., 2013). EC is emitted as primary particles from incomplete combustion sources (i.e., biomass burning and fossil fuel combustion). Unlike EC, OC can either be emitted as primary OC (POC) from combustion sources and non-combustion sources (e.g., biogenic emissions) or formed in the atmosphere as secondary OC (SOC) via the reaction of gas precursors (Hallquist et al., 2009; Jimenez et al., 2009). The sources and abundance of different carbon fractions in carbonaceous aerosols vary considerably in different Chinese cities, as a result of complex interplay between meteorology, local and regional emissions sources, and atmospheric processes (Zhang et al., 2008; Cui et al., 2015; Tie et al., 2017; An et al., 2019). Therefore, quantifying the sources of carbonaceous aerosol in China is a challenging task.

Radiocarbon ([14]C) analysis of carbonaceous aerosols is the most direct and effective method to distinguish their main sources, exploiting the fact that OC and EC of fossil origins (i.e., vehicle emissions, coal combustion) do not contain [14]C (Heal, 2014; Cao et al., 2017; Dusek et al., 2013). [14]C analysis of OC and EC separately provides a clear-cut division of carbonaceous aerosols into four major fractions: fossil OC, non-fossil OC (e.g., OC from biomass burning, biogenic emissions and cooking), fossil EC and biomass-burning EC (e.g., Gustafsson et al., 2009; Szidat et al., 2009; Zotter et al., 2014; Dusek et al., 2017; Ni et al., 2018, 2019a). For example, Liu et al. (2014) demonstrated that fossil sources including coal burning and vehicle emissions dominated EC during winter haze events in Guangzhou, China. Zhang et al. (2015) showed that the elevated carbonaceous aerosols during the severe haze event in January 2013 in China were by a large extent driven by SOC from both fossil and non-fossil precursors. In addition, the analysis of the [13]C/[12]C ratio can refine [14]C source apportionment, because coal combustion and vehicle emissions have different [13]C source signatures although they both completely depleted in [14]C (e.g., Andersson et al., 2015; Li et al., 2016; Winiger et al., 2016, 2017; Fang et al., 2017, 2018; Ni et al., 2018).

A critical question for effective haze mitigation is whether carbonaceous aerosols in different Chinese cities have similar characteristics during haze events. However, there are not many studies highlighting the differences in sources of primary and secondary carbonaceous aerosols between cities, especially for studies employing the analysis of [14]C or the stable isotope [13]C (e.g., Zhang et al., 2015; Andersson et al., 2015; Liu et al., 2016). In this work, we compare the severe haze events reaching

"red alarm" level (i.e., the highest air-quality warning level in China) in two Chinese megacities (Beijing and Xi'an) during December 2016 and January 2017. We present measurements of dual carbon isotopes (i.e., $^{14}$C and the stable carbon isotope $^{13}$C) for EC and OC. The sources of carbonaceous aerosols are elucidated and compared between haze and clean periods in Beijing and Xi'an, with the main objectives: (1) quantitative understanding of the difference in EC contribution from burning of biomass, coal and liquid fossil fuel (i.e., vehicle emissions) under different pollution conditions; and (2) constraint on the

sources of both primary and secondary OC. Furthermore, the comparison of day-time and night-time results in Xi'an yields insight into diurnal variation in sources of carbonaceous aerosols.

## 2 Methods

### 2.1 Aerosol collection

To collect PM$_{2.5}$ samples, high-volume aerosol samplers (flow rate = 1.0 m$^3$ min$^{-1}$; TE-6070 MFC, Tisch Inc., Cleveland, OH,

USA) were used at an urban background site in Xi'an and Beijing (see Table S1 for details about the sampling sites). Xi'an is the largest city in northwestern China, with over 8.8 million residents and 2.5 million vehicles in 2016 (Xi'an Municipal Bureau of Statistics and NBS Survey Office in Xi'an, 2017). Surrounded by Qinling Mountains to the south and the Loess Plateau to the north, days with low wind speed occur frequently in Xi'an, promoting the accumulation of air pollutants. Xi'an is now facing increased serious air quality issues due to the rapid increase of motor vehicles and energy consumption in the past two

decades. Besides residential coal combustion, biomass burning is also a major emission source in Xi'an and its surrounding areas (i.e., Guanzhong Plain) for heating and cooking especially in winter (Zhang et al., 2014; Xu et al., 2016). Beijing, the capital of China, is a megacity with over 21 million residents and 5.7 million vehicles in 2016 (Beijing Municipal Bureau of Statistics and NBS Survey Office in Beijing, 2017). Beijing is located in the Beijing-Tianjin-Hebei region, the most economically developed region in North China. However, the rapid economic growth and urbanization associated with heavy

coal consumption and rapid increase usage of vehicles lead to the poor air quality in Beijing. Besides local emissions, regional transport of pollutants between neighboring cities also contributes to air pollution in Beijing (Zheng et al., 2015; An et al., 2019). The 12 h integrated (daytime: 8:00 a.m. to 8:00 p.m., local standard time, LST; nighttime: 8:00 p.m. to 8:00 a.m. the following day) PM$_{2.5}$ was sampled on pre-combusted quartz filters (8 in × 10 in; QM-A, Whatman Inc., Clifton, NJ, USA) in Xi'an from 1 January 2017 to 10 January 2017. In Beijing, the 24 h integrated (10:00 a.m. to 10:00 a.m. the following day)

PM$_{2.5}$ was collected from 2 December 2016 to 8 January 2017. Field blanks were collected by exposing filters to ambient air for 15 min. Immediately after collection, the filters were transferred into pre-combusted aluminum foils and stored at −18 $^{\circ}$C.

### 2.2 Concentration measurements of OC and EC

The IMPROVE_A protocol (Chow et al., 2007) was implemented on a carbon analyzer (DRI Model 2001, Atmoslytic Inc., USA) for measurement of carbon concentrations. The relative standard deviations for the replicate analyses were smaller than

10 % for OC and EC. OC mass was corrected for field blanks (0.4 μg cm$^{-2}$). EC was too small to be detected on field blanks.

Acidification to remove potential interferences from carbonates is not necessary, because carbonate carbon in $PM_{2.5}$ samples is found to be negligible, compared to the relatively larger OC and EC amounts for both mass determination and carbon isotopic analysis (Supplement S1).

## 2.3 Analysis of carbon isotope

Six samples from haze and clean days were selected per sampling site for carbon isotope analysis (Tables S1 and S2). We define clean days at each site as $PM_{2.5}$ < median concentration in the winter heating season from 15 November 2016 to 15 March 2017. In Xi'an, there were 4 composite samples (2 daytime + 2 nighttime) from haze days, and 2 composite samples (1 daytime + 1 nighttime) from clean days. In Beijing, five 24 h samples were selected from haze days, and 1 composite sample from two clean days. Each composite sample consists of 2 12h (for Xi'an) or 24 h (for Beijing) filter pieces with similar $PM_{2.5}$

loadings that agree within 20 % (Fig. S1).

### 2.3.1 Stable isotope $^{13}C$

Filter samples were placed in a quartz tube with CuO grains. The tube was subsequently evacuated and sealed before heating for 3 h at 375 °C to remove OC. Then the EC was extracted by heating the remaining carbon for 5 h at 850 °C. The $^{13}C/^{12}C$ ratio of EC was measured by an isotope mass spectrometer (Finnigan MAT-251; Bremen, Germany) and expressed in the delta

notation:

$$\delta^{13}C = \left[\frac{(^{13}C/^{12}C)_{sample}}{(^{13}C/^{12}C)_{V-PDB}} - 1\right]. \tag{1}$$

$\delta^{13}C$ values are usually reported in per mil (‰). $(^{13}C/^{12}C)_{V-PDB}$ is the $^{13}C/^{12}C$ ratio of the international standard Vienna Pee Dee Belemnite (V-PDB). A well-characterized standard was measured every working day. Duplicate analysis of $\delta^{13}C$ of EC showed an analytical precision better than ± 0.3‰. This method was detailed in Ni et al., (2019b), where impacts of potential charred

OC on the isolated EC were evaluated using an isotope-mass-balance based sensitivity analysis. We concluded that the expected differences in $\delta^{13}C_{EC}$ are smaller than 1‰ under the assumption that the fraction of charred OC in the isolated EC is at most 20%.

### 2.3.2 Radiocarbon

OC and EC in $PM_{2.5}$ samples were converted to $CO_2$ using an aerosol combustion system (ACS; Dusek et al., 2014). The ACS

has been evaluated in two intercomparison studies (Szidat et al., 2013; Zenker et al., 2017). The isolated $CO_2$ was subsequently reduced to graphite (de Rooij et al., 2010) before $^{14}C$ measurements can be conducted with the accelerator mass spectrometer (AMS) at CIO (van der Plicht et al., 2000). The temperature protocol for OC and EC combustion has been detailed in Ni et al. (2018), and is summarized in Fig. S2. To remove possible interfering gas (e.g., $NO_x$, halogen and water vapor) from $CO_2$, a reduction oven filled with copper grains and silver, a dry ice-ethanol bath and a flask filled with phosphorus pentoxide are

installed on the ACS.

Fraction modern ($F^{14}C$) is used to report the $^{14}C$ data (Reimer et al., 2004). $F^{14}C$ relates the $^{14}C/^{12}C$ ratio of a sample to the ratio of the unperturbed atmosphere in the reference year 1950:

$$F^{14}C = \frac{(^{14}C/^{12}C)_{sample,[-25]}}{(^{14}C/^{12}C)_{1950,[-25]}}.$$
(2)

Both ratios are normalized to $\delta^{13}C$ of -25‰ to remove the effect of isotope fractionation. Practically, $(^{14}C/^{12}C)_{1950, [-25]}$ equals
to the $^{14}C/^{12}C$ ratio of an oxalic acid standard (OXII) multiplied by a factor of 0.7459. Contamination during graphitization and AMS measurements was quantified from the measured $F^{14}C$ of standards (OXII with known $F^{14}C$ of 1.3407 and Rommenhöller with $F^{14}C=0$) processed in the same way as samples. The resulting estimated dead and modern contamination were used to correct the $^{14}C$ data according to Santos et al. (2007). The reliability of data correction was further verified by measuring two secondary standards (i.e., IAEA-C7 and-C8) on the same wheel of samples. The measured values of IAEA-C7 (0.495 ± 0.008) and IAEA-C8 (0.154 ± 0.007) agree with their respective consensus value (0.4953 ± 0.0012 and 0.1503 ± 0.0017) within uncertainties.

## 2.4 Source apportionment methods

$F^{14}C$ is larger than the fraction of non-fossil carbon (i.e., $f_{nf}(OC)$ for OC, $f_{bb}(EC)$ for EC) due to the large release of $^{14}C$ into the atmosphere from the nuclear bomb tests in 1960s. To eliminate this effect, $F^{14}C$ is divided by $F^{14}C$ of non-fossil sources ($F^{14}C_{nf}$). $F^{14}C_{nf}$ is estimated as 1.09 ± 0.05 for OC and 1.10 ± 0.05 for EC (see details in Ni et al., 2019b), using a tree growth model and the contemporary atmospheric $^{14}CO_2$ over the past years (Lewis et al., 2004; Mohn et al., 2008; Levin et al., 2010), with the assumption that biomass-burning OC and biogenic OC contribute to 85% and 15% of total OC, respectively. Once $f_{nf}(OC)$ and $f_{nf}(EC)$ are known, carbon concentrations can be apportioned into EC and OC from non-fossil sources ($EC_{bb}$, $OC_{nf}$) and fossil sources ($EC_{fossil}$, $OC_{fossil}$) (Eq. 3–6 in Table 1). $OC_{nf}$ and $OC_{fossil}$ are further divided into POC from biomass burning ($POC_{bb}$), other non-fossil OC ($OC_{o,nf}$) (Eq. 7–8), primary and secondary fossil OC ($POC_{fossil}$ and $SOC_{fossil}$, respectively; Eq. 9–10). $POC_{bb}$ and $POC_{fossil}$ are estimated using EC as a tracer of primary emissions (i.e., the EC tracer method; Turpin and Huntzicker, 1995). Based on $OC_{o,nf}$ and $SOC_{fossil}$, total SOC and the fraction of fossil carbon in SOC ($f_{fossil}(SOC)$) are estimated using Eq. (11–12). $OC_{o,nf}$ mainly includes SOC of non-fossil origins ($SOC_{nf}$), primary biogenic OC and cooking OC. $OC_{o,nf}$ is approximately $SOC_{nf}$, as contributions of primary biogenic sources and cooking to $OC_{o,nf}$ are likely small (Hu et al., 2010; Guo et al., 2012). If cooking is prominent, $OC_{o,nf}$ is an overestimate of $SOC_{nf}$. To estimate the uncertainties of the source apportionment results, a Monte Carlo simulation (n=10000) using Eq. (3–12) was carried out as described in Supplement S3. The $^{14}C$ source apportionment results are presented in Tables S3 and S4.

The dual carbon isotope signatures of EC were used in a Bayesian Markov chain Monte Carlo (MCMC) scheme (Andersson, 2011), to conduct the mass-balance three source apportionment of EC (e.g., Andersson et al., 2015; Li et al., 2016; Winiger et

al., 2016, 2017; Fang et al., 2017, 2018). That is, the $F^{14}C$ and $\delta^{13}C$ of ambient EC ($F^{14}C_{(EC)}$ and $\delta^{13}C_{EC}$) can be explained by burning of biomass ($_{bb}$), coal ($_{coal}$) and liquid fossil fuel ($_{liq.fossil}$; i.e., vehicle emissions):

$$\begin{pmatrix} F^{14}C_{(EC)} \\ \delta^{13}C_{EC} \\ 1 \end{pmatrix} = \begin{pmatrix} F^{14}C_{nf} & F^{14}C_{coal} & F^{14}C_{liq.fossil} \\ \delta^{13}C_{bb} & \delta^{13}C_{coal} & \delta^{13}C_{liq.fossil} \\ 1 & 1 & 1 \end{pmatrix} \begin{pmatrix} f_{bb} \\ f_{coal} \\ f_{liq.fossil} \end{pmatrix} \tag{13}$$

$F^{14}C_{coal}$ and $F^{14}C_{liq.fossil}$ are equal to zero since coal and liquid fossil fuel do not contain $^{14}C$. $\delta^{13}C_{bb}$, $\delta^{13}C_{coal}$ and $\delta^{13}C_{liq.fossil}$ are $\delta^{13}C$ signatures for EC from the three sources. Their values were established as $\delta^{13}C_{bb}$ (−26.7 ± 1.8 ‰ for C3 plants, and −16.4 ± 1.4 ‰ for corn stalk; mean ± SD), $\delta^{13}C_{coal}$ (−23.4 ± 1.3 ‰) and $\delta^{13}C_{liq.fossil}$ (−25.5 ± 1.3 ‰), based on critical evaluations of literature studies (Andersson et al., 2015; Ni et al., 2018; and references therein). Uncertainties in $F^{14}C$ and $\delta^{13}C$ source signatures and the measured $F^{14}C_{(EC)}$ and $\delta^{13}C_{EC}$ are considered in the MCMC technique (Parnell et al., 2010, 2013). MCMC outputs are the posterior probability density functions for $f_{bb}$, $f_{coal}$ and $f_{liq.fossil}$ (i.e., the relative contribution of each source to EC). The median and interquartile range (25th–75th percentile) are used as the best estimate and the uncertainties, respectively.

## 3 Results and discussion

### 3.1 Fossil and non-fossil contributions to EC and OC

During the measurement periods, the highest daily mass concentrations of PM$_{2.5}$ in Xi'an (~250–420 µg m$^{-3}$) and Beijing (~210–360 µg m$^{-3}$; Fig. S1) were 10–17 and 8–14 times higher than the standard of World Health Organization (25 µg m$^{-3}$; WHO, 2006), respectively. Using radiocarbon measurements, we investigated the sources of carbonaceous aerosols in PM$_{2.5}$ in both cities during several haze periods, and compared them to clean periods, with PM$_{2.5}$ concentrations below 100 µg m$^{-3}$ in Xi'an and below 20 µg m$^{-3}$ in Beijing. In Xi'an, even during clean periods we defined here, daily PM$_{2.5}$ concentrations were higher than the Chinese pollution standard of 75 µg m$^{-3}$, reflecting severe air quality problems. PM$_{2.5}$, OC and EC concentrations during haze periods were > 2 times higher in Xi'an and > 5 times higher in Beijing than those during clean periods, respectively. OC/EC ratios in Xi'an slightly decreased from ~4 during haze periods to ~3 during clean periods, while OC/EC ratios in Beijing were lower during haze periods (~3) than clean periods (~4). This reflects different sources and formation mechanisms of haze pollution in the two cities. In Xi'an, we collected day and night PM$_{2.5}$ samples. No consistent day-night variations in concentrations of PM$_{2.5}$, OC and EC (Figs. 1 and S1) were observed. This results from the diurnal cycle of human activities (e.g., traffic, usage of biomass and coal for heating or cooking) and the development of planetary boundary layer height which controls the vertical mixing and dilution of pollutants.

Radiocarbon ($^{14}C$) in EC and OC was measured to distinguish their fossil (mainly coal burning and traffic emissions) and non-fossil sources (mainly biomass burning). The most important contributor to EC was fossil fuel combustion, both in Xi'an and Beijing, contributing 73 ± 2% in Xi'an and 80 ± 3% in Beijing. The remaining EC arose from biomass burning (27 ± 2% in Xi'an and 20 ± 3% in Beijing; Fig. 1). In Xi'an, the fraction of biomass-burning EC in total EC ($f_{bb}(EC)$) was largely constant

during haze and clean periods (range: 25%–29%), regardless of the wide concentration range of EC from biomass burning (EC$_{bb}$, 1.8–6.4 μg m$^{-3}$) and fossil fuel combustion (EC$_{fossil}$, 4.3–18 μg m$^{-3}$). This suggests that the increase in EC$_{fossil}$ and EC$_{bb}$ concentrations during haze periods in Xi'an is likely caused by the enhanced emissions from both fossil fuel and biomass burning by a similar factor and due to meteorological conditions favoring the accumulation of particulate air pollutions. $f_{bb}$(EC) values in Beijing (20 ± 3% with a range of 17%–24%) were consistently smaller than those in Xi'an (range: 25%–29%), showing that fossil sources contribute more strongly to EC in Beijing. Moreover, during haze periods in Beijing, $f_{bb}$(EC) increased with increasing total EC concentrations (Fig. 2).

In Xi'an, OC concentrations from non-fossil sources averaged 29 ± 16 μg m$^{-3}$ (OC$_{nf}$; range: 9–49 μg m$^{-3}$), slightly higher than those from fossil sources (OC$_{fossil}$; 24 ± 13 μg m$^{-3}$; range: 8–40 μg m$^{-3}$) at 95% confidence level (paired $t$ test, $p$-value = 0.01). However, in Beijing, OC$_{nf}$ (12 ± 5 μg m$^{-3}$; 3–19 μg m$^{-3}$) was significantly lower than OC$_{fossil}$ (24 ± 10 μg m$^{-3}$; 4–33 μg m$^{-3}$) ($p$-value = 0.001). Consequently, the relative contribution of OC$_{nf}$ to total OC ($f_{nf}$(OC)) was much higher in Xi'an (average 54 ± 2 %) than in Beijing (34 ± 3%). $f_{nf}$(OC) in both cities was considerably higher than the corresponding $f_{bb}$(EC) for all samples (Fig. 1). The main reason for larger $f_{nf}$(OC) than $f_{bb}$(EC) is that primary OC/EC ratios from biomass burning emissions are higher than those from fossil sources. So even though biomass burning contributes a small portion of EC, its contribution to primary OC will be much higher. In addition, other non-combustion sources (e.g., biogenic emissions, cooking fumes) and secondary formation contribute only to OC, but not to EC.

In this study, the $f_{fossil}$(EC) values in Xi'an during winter 2016/2017 are comparable with those previously measured during winter 2015/2016 and winter 2008/2009 (Ni et al., 2018, 2019b), as illustrated in Fig. 3b, pointing to relative constant contribution of fossil fuel combustion vs. biomass burning to EC in Xi'an over the past decade. As shown in Fig. 3b, the $f_{fossil}$(EC) values in Beijing during winter 2016/2017 agree with the values reported at an urban site of Beijing in January 2014 (Fang et al., 2017). A slightly higher $f_{fossil}$(EC) in urban Beijing was observed during February 2010 (Chen et al., 2013). Despite the slight variation of $f_{fossil}$(EC) over time, $f_{fossil}$(EC) in Beijing is generally higher than that in Xi'an (Fig. 3b). The presented overall average $f_{fossil}$(OC) for winter 2016/2017 in Beijing (66 ± 3%) was higher than that in Xi'an (46 ± 2%), consistent with previously reported $f_{fossil}$(OC) in Beijing and Xi'an (Zhang et al., 2015; Ni et al., 2019a). Lower $f_{fossil}$(OC) values in winter were reported for Chongqing (24%), and higher $f_{fossil}$(OC) was observed in Taiyuan (71%) during winter 2013/2014 (Ni et al., 2019a). The comparison of $f_{fossil}$(EC) and $f_{fossil}$(OC) in different Chinese cities indicates that the relative importance of fossil sources in carbonaceous aerosols varies spatially, and can change over the years. In Xi'an, clean periods showed a slight day-night difference with increased contributions of fossil sources to EC and OC during the day. During haze periods, especially the 2$^{nd}$ haze event (XH_day2, XH_night2), this day-night difference disappeared, which suggests a long residence time of the pollution particles in the urban atmosphere during haze events.

Overall, our [14]C data show that fossil sources contribute more strongly to EC and OC in Beijing than in Xi'an, which is consistent with previous observations. Both in Beijing and Xi'an the fossil vs. non-fossil contributions to EC and OC did not

change drastically during haze and clean periods. In Xi'an, a slight day-night difference was observed during clean periods, but disappeared during haze periods, suggesting a large accumulation of particles.

## 3.2 Fossil EC apportioned by stable carbon isotopes: coal vs. liquid fossil fuel

Besides $F^{14}C$, the $\delta^{13}C$ of EC adds additional dimension where fossil EC can be distinguished into EC from burning of coal and of liquid fossil fuel (i.e., vehicle emissions). Considerable geographical differences in $\delta^{13}C_{EC}$ signatures were observed, with more depleted values in Xi'an (−25.1 ± 0.5‰; −25.6‰ to −24.4 ‰) relative to those in Beijing (−24.1 ± 0.4‰; −24.4‰ to −23.4‰; Fig. 3). The Xi'an signatures are consistent with the signature of liquid fossil fuel combustion ($\delta^{13}C_{liq.fossil}$ = −25.5 ± 1.3‰; Sect. 2.4), whereas the more enriched values in Beijing indicate the influence of coal combustion ($\delta^{13}C_{coal}$ = −23.4 ± 1.3‰).

In both Xi'an and Beijing, moderate differences exist in $\delta^{13}C_{EC}$ between clean and haze days, pointing to a shift in combustion sources. In Xi'an, $\delta^{13}C_{EC}$ during clean periods (~−25.5‰) was slightly depleted compared to that during haze periods (−25.0‰ to −24.4‰), whereas Beijing exhibited more enriched $\delta^{13}C_{EC}$ during clean periods (−23.4‰) than during haze periods (−24.4‰ to −24.1‰). This suggests a moderate increase in coal combustion contribution to EC in Xi'an during haze days and a decrease in Beijing. In Xi'an, no strong day-night difference in $\delta^{13}C_{EC}$ was observed, with the largest absolute differences of 0.5‰ between XH_day1 and XH_night1. The day-night differences are small relative to the uncertainties of the potential sources; for example, the endmember range for coal combustion is more uncertain (± 1.3‰). The small day-night differences in $\delta^{13}C_{EC}$ reflect well-mixed EC emissions.

The Bayesian MCMC model takes into account the uncertainties of the $\delta^{13}C$ and $F^{14}C$ endmembers and statistically apportions EC into the fraction of biomass burning ($f_{bb}$), coal combustion ($f_{coal}$) and liquid fossil fuel combustion ($f_{liq.fossil}$). The MCMC-derived $f_{bb}$ is in principle the same as the $^{14}C$-based $f_{bb}$(EC) (Fig. S3). The MCMC results (Fig. 4) show that there were no strong day-night differences in EC sources during haze and clean periods in Xi'an. Liquid fossil fuel combustion was the most important contributor to EC in Xi'an, with increased contribution during clean periods. In Beijing, coal combustion was the dominant source of EC, with the relative contribution ranging from 48% (median; 31%−61%, interquartile range) during haze periods to 61% (45%−71%) during clean periods. $f_{bb}$ was fairly constant between haze and clean periods with respect to $f_{coal}$ and $f_{liq.fossil}$ for all samples. In Xi'an, $f_{bb}$ was comparable to $f_{coal}$ during haze days, and larger than $f_{coal}$ during clean days. In Beijing, biomass-burning EC was the smallest fraction in total EC, with smaller $f_{bb}$ than $f_{coal}$ during both haze and clean days. Concentrations of total EC increased by 2 times from clean days (~7.4 μg m$^{-3}$) to haze days (18.0 μg m$^{-3}$) in Xian, and 8 times in Beijing (1.6 μg m$^{-3}$ to 13.5 μg m$^{-3}$). The increased EC concentrations during haze periods in Xi'an were attributed to liquid fossil fuel combustion (43%), coal combustion (29%) and biomass burning (28%). However, in Beijing, coal combustion contributed most of the increased concentrations of EC (45%), followed by burning of liquid fossil fuel (33%) and biomass (22%).

In summary, complementing $^{14}C$ with $\delta^{13}C$ allows for quantitative constraints on EC sources: EC was dominated by liquid fossil fuel combustion (i.e., vehicle emissions) in Xi'an and by coal burning in Beijing, especially during clean periods. In Xi'an, no strong day-night differences in EC sources were observed during haze and clean periods. Compared with earlier observations in Xi'an (Fig. 3b), we found that the $\delta^{13}C_{EC}$ values in January 2017 from this study are comparable with wintertime $\delta^{13}C_{EC}$ in 2015/2016 (Ni et al., 2019b), but much more depleted than wintertime $\delta^{13}C_{EC}$ in 2008/2009 (Ni et al., 2018) and January 2003 (Cao et al., 2011). This suggests that fossil sources of EC in Xi'an have changed in the past decade, with decreasing relative contribution from coal combustion. This is in line with recent changes in energy use, and the decreasing enrichment factors of As and Pb (i.e., indicators of coal combustion) in Xi'an, as documented in recent studies (Xu et al., 2016).

As shown in Fig. 3b, in Beijing, variations in $\delta^{13}C_{EC}$ from January 2003 (Cao et al., 2011) to January 2017 (this study) are much narrower than those in Xi'an, indicating that EC combustion sources did not change significantly throughout the years in Beijing. Our $\delta^{13}C_{EC}$ values overlap with those in January 2014 (Fang et al., 2017) and fall into the range of reported $\delta^{13}C_{EC}$ values in urban Beijing (Cao et al., 2011; Chen et al., 2013) and the regional receptor site of Beijing (Andersson et al., 2015; Fang et al., 2017).

## 3.3 Primary and secondary OC

As explained in Sect. 2.4, $OC_{nf}$ and $OC_{fossil}$ are apportioned into primary ($POC_{bb}$, $POC_{fossil}$) and secondary OC ($OC_{o,nf}$, $SOC_{fossil}$; Fig. 5). The large error bars of their concentrations reflect the large uncertainties in $r_{bb}$ and $r_{fossil}$. It should be noticed that $OC_{o,nf}$ is used as an approximation of $SOC_{nf}$, or can be regarded as an upper limit of $SOC_{nf}$ if cooking is a prominent OC source.

In Xi'an, both ratios of $OC_{o,nf}/POC_{bb}$ and $SOC_{fossil}/POC_{fossil}$ increased during haze periods (Fig. 5a). $OC_{o,nf}/POC_{bb}$ ratio increased by 2.5 times from 0.33–0.46 during clean periods to 0.86–1.1 during haze periods; in contrast $SOC_{fossil}/POC_{fossil}$ increased by 1.5 times from 0.46–0.50 to 0.62–0.78. This underlines that haze episodes in Xi'an were mainly caused by additional SOC formation, with larger contribution from non-fossil sources than fossil sources. As shown in Fig. 5b, the contribution of SOC (i.e., $SOC \cong OC_{o,nf} + SOC_{fossil}$) to OC increased from clean periods (28%–32%) to haze periods (44%–48%); this mainly resulted from increased contribution of $OC_{o,nf}$ to total OC (i.e., from 14%–16% to 26%–29%). In Xi'an, the day-night difference was larger during clean periods with less SOC at night for both absolute concentration and relative contribution to total OC (Figs. 5b, 5c).

In contrast, Beijing had the opposite variation trends of $OC_{o,nf}/POC_{bb}$ and $SOC_{fossil}/POC_{fossil}$ from clean to haze periods. $OC_{o,nf}/POC_{bb}$ ratios during clean periods (1.3) were on average five times higher than those during haze periods (0.18–0.33), and $SOC_{fossil}/POC_{fossil}$ ratios during clean periods (0.71) were slightly higher than those during haze periods (0.41–0.64). This suggests that in Beijing the increased OC concentrations during haze periods were mainly derived from elevated concentrations of $POC_{bb}$ and $POC_{fossil}$. As shown in Fig. 5b, high SOC contribution to total OC was observed during clean periods, mainly due to elevated contribution from $OC_{o,nf}$. The $OC_{o,nf}$ is not likely attributed to biogenic OC, because the biogenic emissions are very low in winter. As a result, the elevated contribution from $OC_{o,nf}$ to OC during clean periods in Beijing could be attributed

to regional sources. During clean periods, concentrations of OC and $OC_{o,nf}$ are small, and the measured carbon concentrations can reflect regional sources, which are dominated by secondary sources due to long-range transport. It could also be that contribution of cooking OC to $OC_{o,nf}$ can be noticeable during clean conditions.

The fossil fraction of the total SOC can be defined as $f_{fossil}(SOC) = SOC_{fossil}/SOC$. In Xi'an around half of SOC was derived from fossil sources ($f_{fossil}(SOC) = 44 \pm 6\%$), whereas $f_{fossil}(SOC) = 75 \pm 10\%$ in Beijing. Using a similar approach with this study, Zhang et al. (2015) also found that Beijing had higher $f_{fossil}(SOC)$ (48%−63%) than in Xi'an (30%–35%) during the January 2013 severe haze events. These findings suggest the important contribution of fossil sources to SOC in Beijing and non-fossil sources in Xi'an. $f_{fossil}(SOC)$ in Beijing increased during haze periods, whereas the opposite trend was found in Xi'an (Fig. 6). During haze periods in Beijing, $f_{fossil}(SOC)$ overlapped with $f_{fossil}(EC)$, and was clearly higher than $f_{fossil}(OC)$.

Together these results reveal the differences in primary and secondary OC in two Chinese megacities. The contribution of SOC to total OC increased from clean to haze periods in Xi'an. In contrary, SOC/OC ratios increased from haze to clean periods in Beijing, mainly due to increased SOC from non-fossil sources. SOC was dominated by fossil sources in Beijing but by non-fossil sources in Xi'an, especially during haze periods. In Xi'an, the day-night difference was larger during clean periods with less SOC at night.

### 3.4 Differences between the fractions of non-fossil carbon in OC and EC

The differences between $f_{nf}(OC)$ and $f_{bb}(EC)$ were smaller in Beijing, ranging from 11% to 20%, compared to 25%–29% in Xi'an. To better understand what governs the differences, we express $f_{nf}(OC)$ in terms of fossil to biomass burning ratio in EC and primary OC/EC emissions ratios. Starting from the formulas of $f_{bb}(EC)$ and $f_{nf}(OC)$:

$$f_{bb}(EC) = \frac{EC_{bb}}{EC_{bb} + EC_{fossil}} = \frac{1}{1 + \frac{EC_{fossil}}{EC_{bb}}} \tag{14}$$

$$f_{nf}(OC) = \frac{OC_{nf}}{OC_{nf} + OC_{fossil}} = \frac{POC_{bb} + OC_{o,nf}}{POC_{bb} + OC_{o,nf} + POC_{fossil} + SOC_{fossil}} \tag{15}$$

We find that

$$f_{nf}(OC) = \frac{1}{1 + \frac{(1 + SOC_{fossil}/POC_{fossil})}{(1 + OC_{o,nf}/POC_{bb})} \times \frac{POC_{fossil}}{POC_{bb}}} =$$

$$\frac{1}{1 + \frac{(1 + SOC_{fossil}/POC_{fossil})}{(1 + OC_{o,nf}/POC_{bb})} \times \frac{r_{fossil}}{r_{bb}} \times \frac{EC_{fossil}}{EC_{bb}}} \tag{16}$$

where $r_{fossil}$ is the weighted average of $r_{coal}$ and $r_{vehicle}$.

Comparing Eq. (14) with Eq. (16), we find that $f_{nf}(OC)$ and $f_{bb}(EC)$ would be equal if $\frac{(1 + SOC_{fossil}/POC_{fossil})}{(1 + OC_{o,nf}/POC_{bb})} \times \frac{r_{fossil}}{r_{bb}} = 1$. Since

$r_{fossil}$ is usually smaller than $r_{bb}$, $f_{nf}(OC)$ tends to be larger than $f_{bb}(EC)$, assuming that SOC formation is comparable for fossil or non-fossil sources (i.e., $\frac{(1+SOC_{fossil}/POC_{fossil})}{(1+OC_{o,nf}/POC_{bb})} \sim 1$). With smaller $r_{fossil}$ than $r_{bb}$, similar $f_{nf}(OC)$ and $f_{bb}(EC)$ can result from larger secondary formation from fossil sources than non-fossil sources (i.e., $\frac{(1+SOC_{fossil}/POC_{fossil})}{(1+OC_{o,nf}/POC_{bb})} > 1$). However, the fossil source coal combustion has a higher primary OC to EC ratio than vehicle emissions (i.e., $r_{coal} > r_{vehicle}$). Therefore, in a city where biomass burning and coal combustion are the dominant pollution sources, $f_{nf}(OC)$ and $f_{bb}(EC)$ will be more similar than in a city where the main sources are biomass burning and vehicle emissions.

Compared to Xi'an, Beijing had significantly smaller differences between $f_{bb}(EC)$ and $f_{nf}(OC)$ (Fig. 1), which was also observed in previous studies during the haze event in January 2013 (Zhang et al., 2015). Comparing Eq. (14) with Eq. (16), this suggests either strong contribution from coal combustion in Beijing or large secondary formation from fossil sources, or both. The stronger contribution of coal combustion to OC in Beijing than in Xi'an was a direct consequence of a larger proportion of coal combustion in EC in Beijing, as demonstrated by the Bayesian MCMC results of EC (Sect. 3.2). The latter was further validated by the variation of SOC. The $f_{fossil}(SOC)$ in Beijing was higher than that in Xi'an, despite the variations between haze and clean periods (Sect. 3.3). By combining [14]C measurements with other state-of-art analytical techniques (e.g., aerosol mass spectrometry), Huang et al. (2014) also found that fossil OC was mostly secondary in nature in Beijing, and non-fossil SOC formation was dominant in Xi'an during a wintertime haze episode (i.e., Beijing had a larger $f_{fossil}(SOC)$ than Xi'an). However, atmospheric mechanisms responsible for the enhancement in fossil-derived secondary organic aerosol formation in Beijing remain unclear.

Furthermore, as shown in Fig. 1, unlike Xi'an where the differences between $f_{nf}(OC)$ and $f_{bb}(EC)$ were relatively constant for all samples, in Beijing the differences between $f_{nf}(OC)$ and $f_{bb}(EC)$ were smaller during haze periods than clean periods, caused by decreased $f_{bb}(EC)$ and slightly increased $f_{nf}(OC)$ during clean periods. This might indicate a higher relative contribution from coal combustion and/or fossil-dominated SOC during haze periods in Beijing. However, the Bayesian MCMC results of EC show the opposite, i.e., in Beijing the contribution of coal combustion to EC was lower during haze periods than during clean periods (Sect. 3.2). Therefore, the only possible explanation is that, during haze periods in Beijing, SOC was dominated by fossil sources. This is validated by significantly larger $f_{fossil}(SOC)$ during haze periods (76%−81%) than during clean periods (~55%; Sect. 3.3).

In conclusion, this section discusses the factors that govern the differences between $f_{bb}(EC)$ and $f_{nf}(OC)$, and concludes that smaller differences suggest stronger contribution from coal combustion and/or larger secondary formation from fossil sources. This is further examined and validated by source apportionment results of EC and OC (Sect. 3.1–3.3).

**4 Conclusion**

In this study the sources of carbonaceous aerosol were quantified using a dual-carbon isotopic approach for $PM_{2.5}$ samples collected in urban Xi'an and Beijing reaching "red alarm" level during December 2016 and January 2017. The $^{14}C$ results showed that fossil sources dominated EC, contributing on average $73 \pm 2$ % of EC in Xi'an and $80 \pm 3$% of EC in Beijing. The

remaining EC was attributed to biomass burning. In Xi'an, $f_{bb}$(EC) was fairly constant during haze and clean periods, despite the wide range of EC concentrations. However, in Beijing, $f_{bb}$(EC) increased with increasing EC concentrations. Complementing $^{14}C$ with $\delta^{13}C$ in a Bayesian MCMC approach allows for separation of fossil sources of EC into coal combustion and liquid fossil fuel combustion. The MCMC results in Xi'an suggest that liquid fossil fuel combustion contributed 44%–49% of EC during haze periods, and 54%–57% of EC during clean periods. In Beijing, coal combustion was

the dominating fossil source of EC, with decreasing contribution to EC from clean periods (~61%) to haze periods (~48%).

$^{14}C$ measurements of OC showed that the contribution of non-fossil sources to OC was larger than that to EC, and was on average $54 \pm 2$ % in Xi'an and $34 \pm 3$% in Beijing. The differences between non-fossil fraction in OC and EC were smaller in Beijing and larger in Xi'an. In Xi'an, the fraction of SOC in total OC was larger during haze periods than during clean periods, mainly due to increased SOC from non-fossil sources. Beijing showed the opposite trends with a larger fraction of SOC in

total OC during clean periods than during haze periods, mainly due to elevated contribution from non-fossil SOC during clean periods.

SOC was dominated by non-fossil sources in Xi'an but by fossil sources in Beijing, especially during haze periods. The relative contribution of fossil sources to SOC ($f_{fossil}$(SOC)) was consistently higher in Beijing than in Xi'an. In Beijing, $f_{fossil}$(SOC) was higher during haze periods (76%–81%) than during clean periods (55%), whereas an opposite trend was found in Xi'an,

with $f_{fossil}$(SOC) increasing from ~39%–43% during haze periods to ~52% during clean periods. In Xi'an, a slight day-night difference was found during clean periods, with increasing fossil contribution to OC and EC during the day and less SOC at night. During strong haze, this day-night difference was negligible, suggesting a large accumulation under stagnant weather conditions during the severe haze periods.

*Data availability.* Data used to support the findings in this study are archived at the Institute of Earth Environment, Chinese
Academy of Sciences, and are available on request from the corresponding author.

*Competing interests.* The authors declare that they have no conflict of interest.

*Author contributions.* RJH and UD designed the study. Isotope measurements were made by HN, MMC, and JG. Data analysis
and interpretation were made by HN, MMC, RJH, and UD. HN wrote the paper with contributions from all co-authors.

*Acknowledgments.* We acknowledge the financial support from the Gratama Foundation. We thank Marc Bleeker and Henk
Been for their help with the [14]C measurements. Special thanks are also given to Anita Aerts-Bijma and Dicky van Zonneveld
for their assistance in [14]C data correction at CIO.

*Financial support.* The authors acknowledge the project grants from the National Key Research and Development Program of
China (no. 2017YFC0212701), the National Natural Science Foundation of China (nos. 41925015, 91644219, and 41877408),
the KNAW (no. 530-5CDP30), the Chinese Academy of Sciences (nos. ZDBS-LY-DQC001), and the Cross Innovative Team
fund from the State Key Laboratory of Loess and Quaternary Geology (no. SKLLQGTD1801).

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

**Table 1.** Equations for $^{14}C$ source apportionment of EC and OC. See Sect. 2.4 for details. $r_{bb}$ and $r_{fossil}$ are primary OC/EC ratio for biomass burning and fossil fuel combustion, respectively. Estimation of $r_{bb}$ and $r_{fossil}$ is presented in Supplement S2.

| Equations | |
|---|---|
| $EC_{bb} = EC \times f_{bb}(EC)$ | (3) |
| $EC_{fossil} = EC \times (1 - f_{bb}(EC)) = EC \times f_{fossil}(EC)$ | (4) |
| $OC_{nf} = OC \times f_{nf}(OC)$ | (5) |
| $OC_{fossil} = OC \times (1 - f_{nf}(OC)) = OC \times f_{fossil}(OC)$ | (6) |
| $POC_{bb} = EC_{bb} \times r_{bb}$ | (7) |
| $OC_{o,nf} = OC_{nf} - POC_{bb}$ | (8) |
| $POC_{fossil} = EC_{fossil} \times r_{fossil}$ | (9) |
| $SOC_{fossil} = OC_{fossil} - POC_{fossil}$ | (10) |
| $SOC = SOC_{nf} + SOC_{fossil} \cong OC_{o,nf} + SOC_{fossil}$ | (11) |
| $f_{fossil}(SOC) = SOC_{fossil}/SOC$ | (12) |

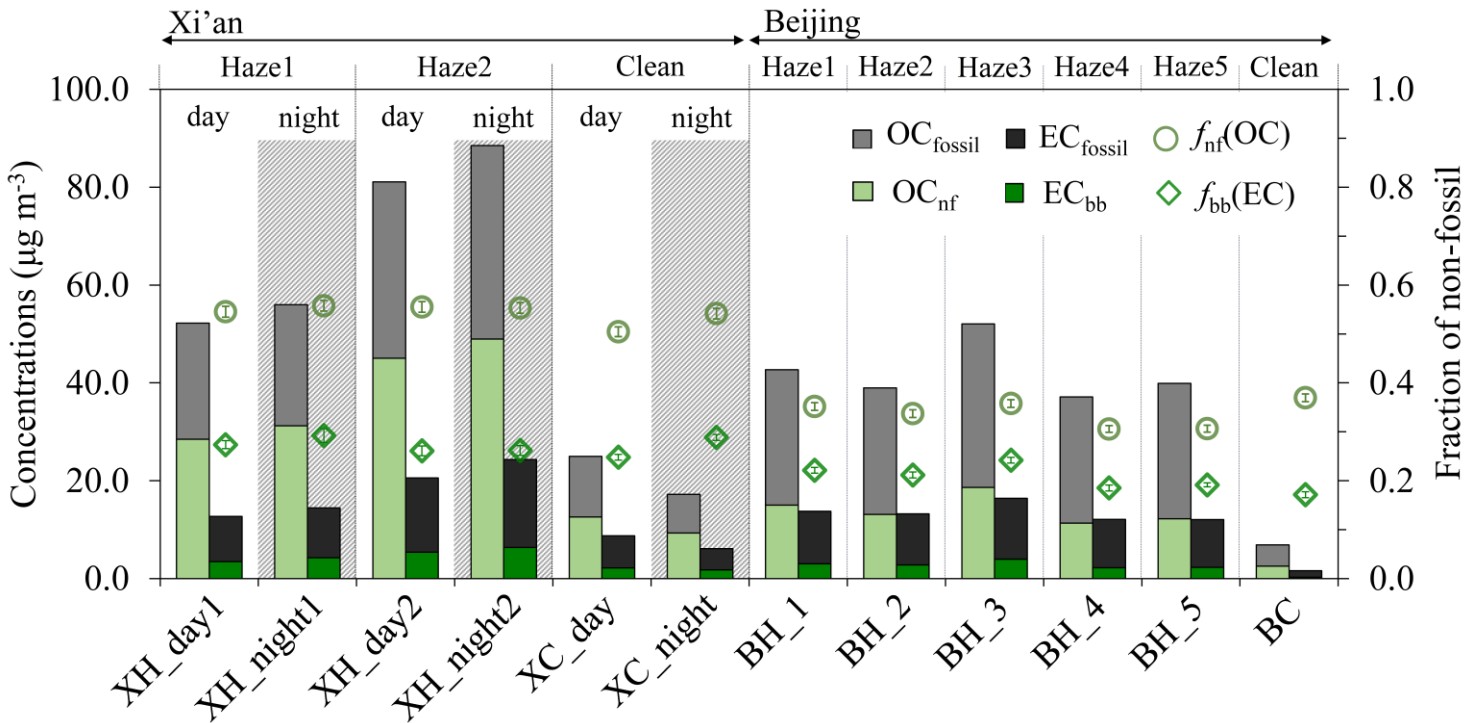

**Figure 1.** Mass concentrations of OC and EC from fossil and non-fossil sources (OC$_{fossil}$, OC$_{nf}$, EC$_{fossil}$ and EC$_{bb}$) as well as fraction of non-fossil carbon in OC and EC ($f_{nf}$(OC) and $f_{bb}$(EC), respectively) for daytime and nighttime PM$_{2.5}$ samples in Xi'an, and 24h-integrated PM$_{2.5}$ samples in Beijing during haze and clean periods during the measurement periods (2 December 2016 to 10 January 2017). For each city "haze" and "clean" are used to represent high and low pollution events, and clean days at each site are defined as days with PM$_{2.5}$ < median concentration in the winter heating season from 15 November 2016 to 15 March 2017. Uncertainties of [14]C-apportioned $f_{nf}$(OC) and $f_{bb}$(EC) are indicated but are too small to be visible. The data are shown in Table S3.

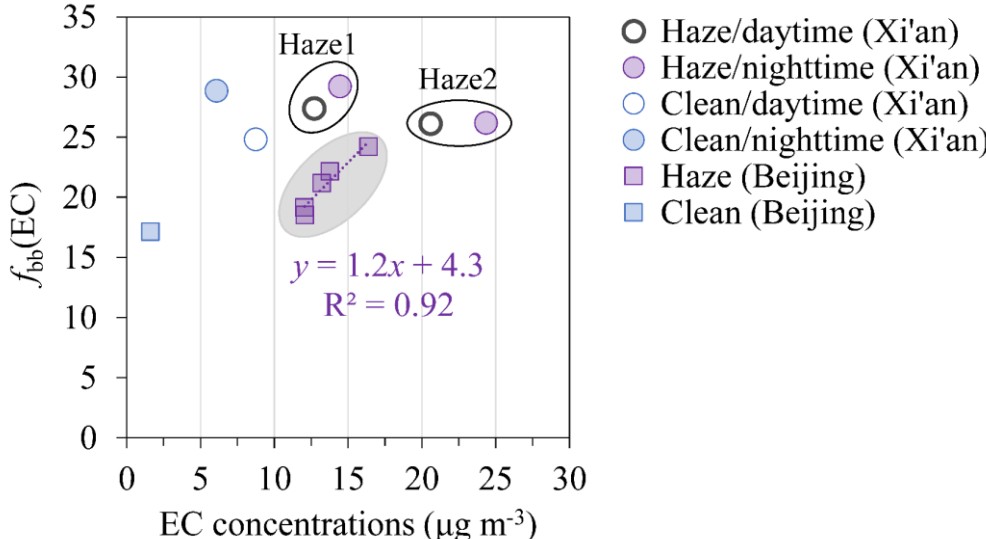

**Figure 2.** Scatter plot of the relative contribution of biomass burning to EC ($f_{bb}$(EC); %) against EC concentrations.

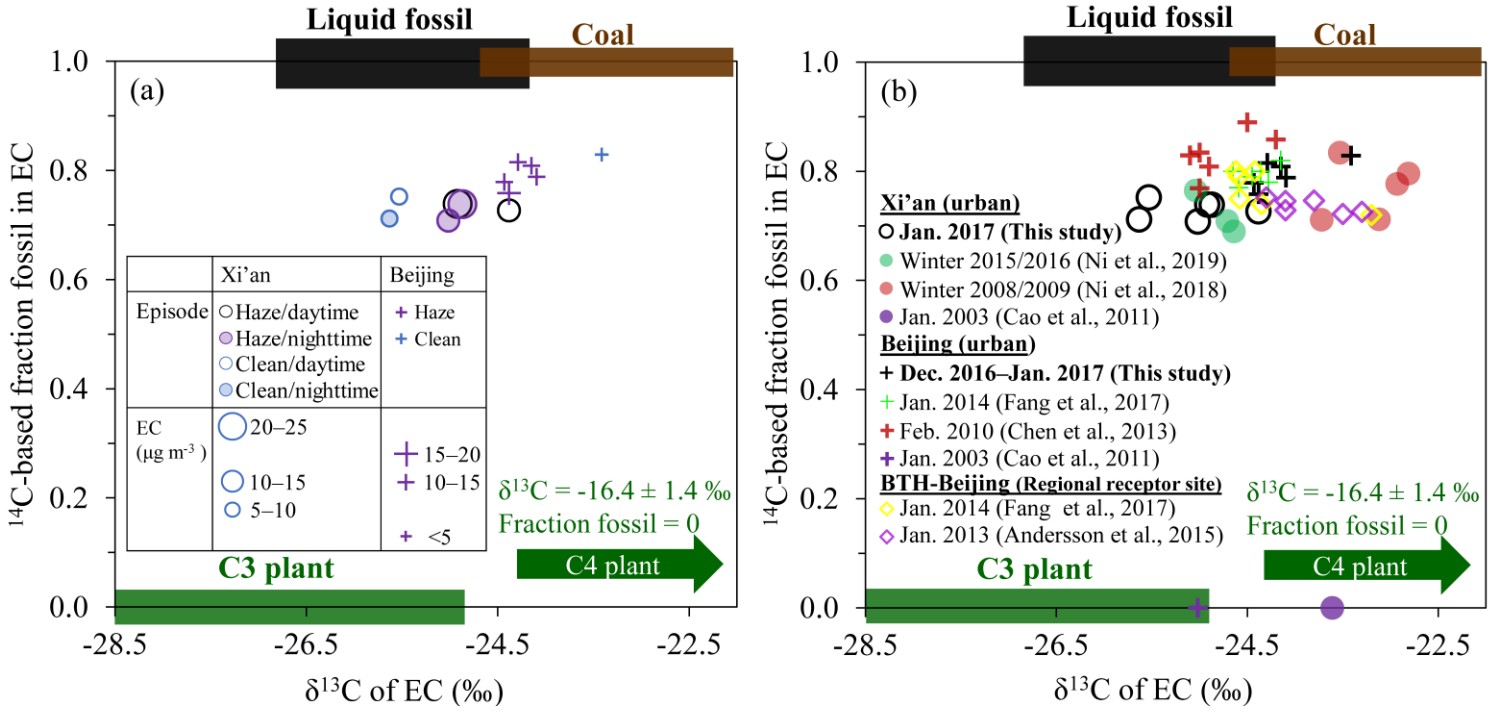

**Figure 3. (a)** [14]C-based fraction fossil versus $\delta^{13}C$ for EC during haze and clean periods in Xi'an and Beijing, China. The symbol size is an indicator of EC concentrations. **(b)** Comparison with previous observations in Xi'an and Beijing, where BTH-Beijing is a regional receptor site of Beijing, located at 100 km southwest of Beijing. Samples from Cao et al. (2011) are placed on the *x*-axis, because no [14]C data were available. The expected [14]C and $\delta^{13}C$ endmember ranges for emissions from C3 plant burning, liquid fossil fuel burning and coal burning are shown as green, black and brown bars, respectively. The $\delta^{13}C$ source signatures are indicated as mean ± SD (Sect. 2.4). The $\delta^{13}C$ signature of corn stalk burning (i.e., C4 plant; −16.4 ± 1.4 ‰) is also indicated.

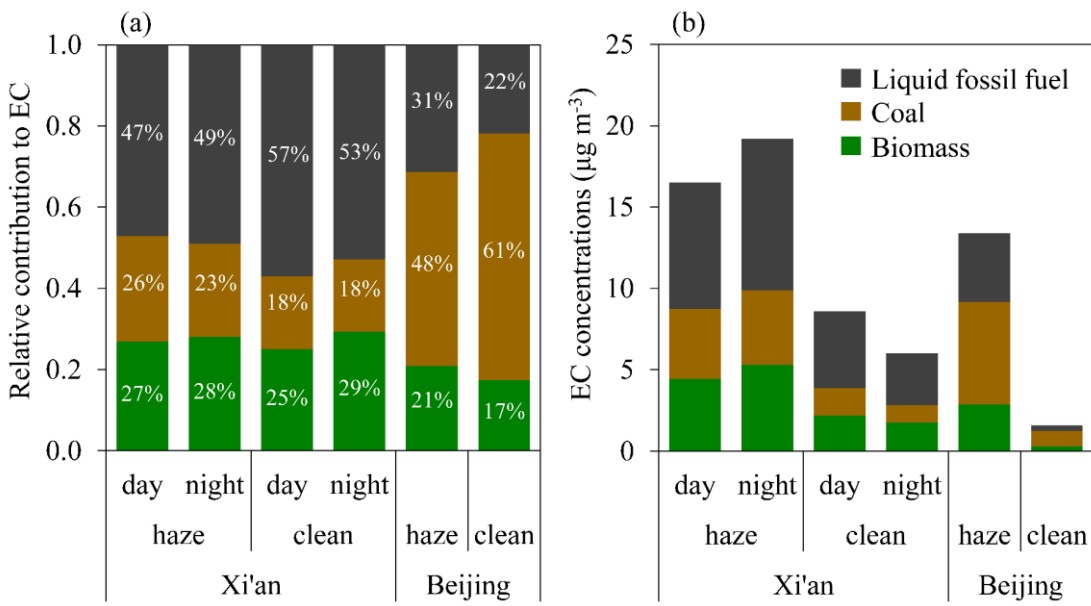

**Figure 4**. **(a)** Fractional contributions of three combustion sources to EC during haze and clean periods in Xi'an and Beijing. **(b)** EC concentrations (µg m$^{-3}$) from each combustion source. The data are presented in Tables S5 and S6.

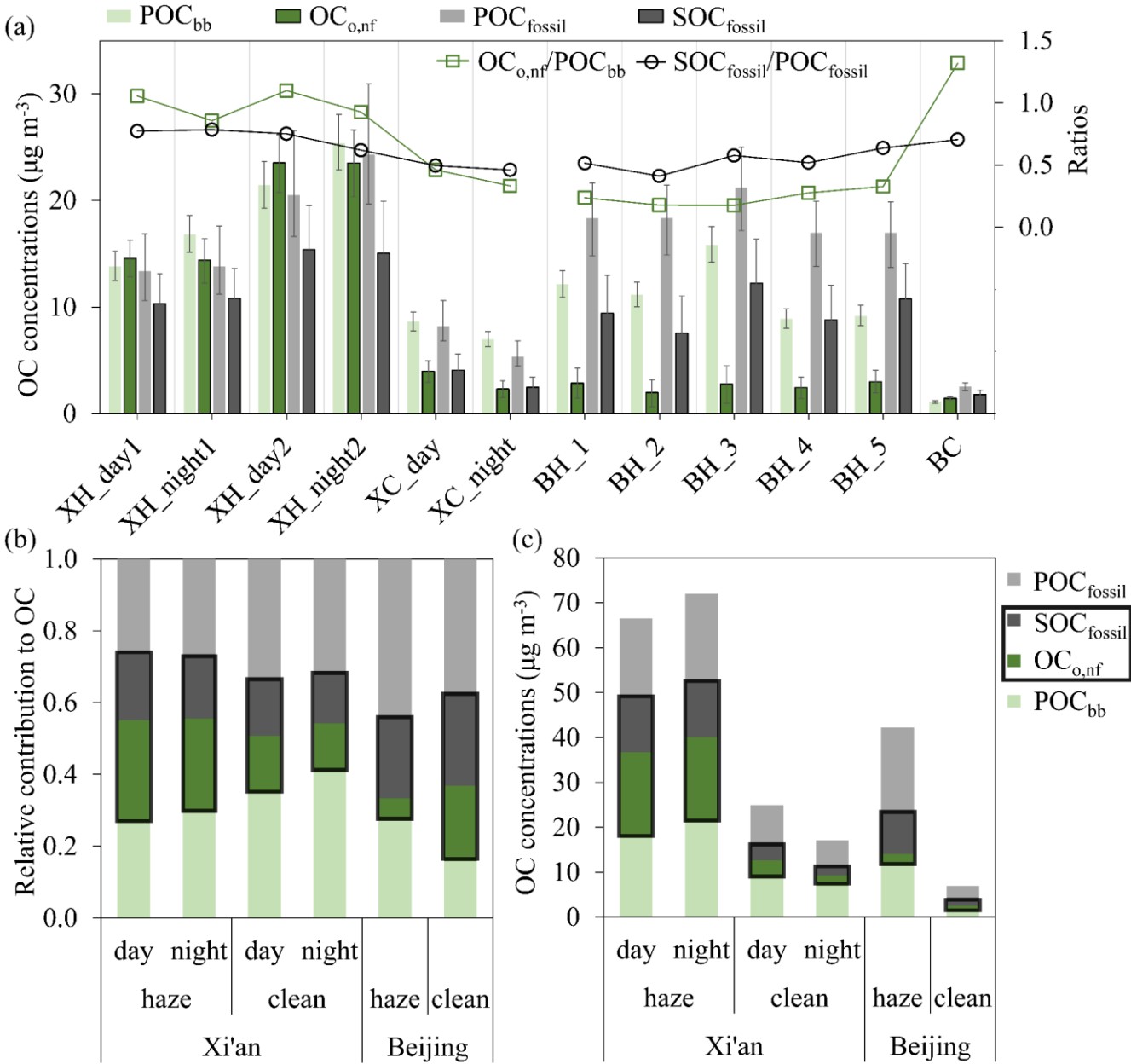

550

**Figure 5. (a)** Concentrations of POC$_{bb}$, OC$_{o,nf}$, POC$_{fossil}$ and SOC$_{fossil}$ (μg m$^{-3}$), and the mass ratio of OC$_{o,nf}$/POC$_{bb}$ and SOC$_{fossil}$/POC$_{fossil}$ of each sample. The error bars indicate the interquartile range (25th–75th percentile) of the median concentrations. Averaged fraction **(b)** and concentration **(c)** of POC$_{bb}$, OC$_{o,nf}$, POC$_{fossil}$ and SOC$_{fossil}$ in total OC during haze and clean periods in Xi'an and Beijing, China. The data are given in Table S4.

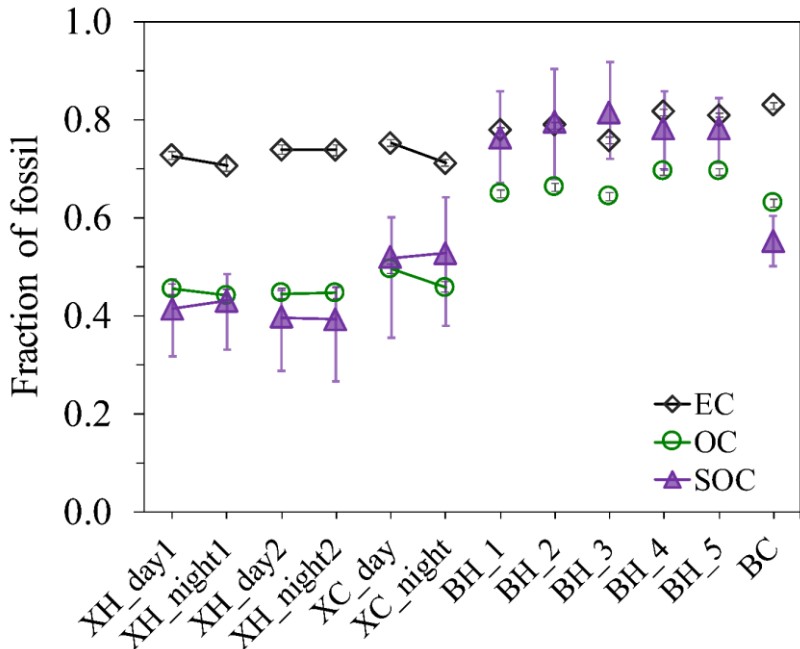

555

**Figure 6.** Fraction of fossil carbon in EC, OC and SOC ($f_{fossil}$(EC), $f_{fossil}$(OC) and $f_{fossil}$(SOC), respectively) during haze and clean periods in Xi'an and Beijing. Interquartile ranges (25th–75th percentile) of the median $f_{fossil}$(SOC) are shown as vertical bars in purple. Uncertainties of [14]C-apportioned $f_{fossil}$(EC) and $f_{fossil}$(OC) are indicated but are too small to be visible. The studied samples include day and night samples in Xi'an (X) and 24 h integrated samples in Beijing (B) during haze periods 560 ("H" samples) and clean periods ("C" samples). For details of each sample (x-axis), see Fig. 1 and Table S1.