# Peer review of "Measurement report: Dual-carbon isotopic characterization of carbonaceous aerosol reveals different primary and secondary sources in Beijing and Xi'an during severe haze events"

_Atmospheric Chemistry and Physics, 2020_

## Referee Comment (RC1) · Anonymous Referee #2 · 20 Jul 2020

The overall quality

The article by Ni et al entitled "Measurement report: Dual-carbon isotopic character-ization of carbonaceous aerosol in Beijing and Xi'an: distinctions in primary versus secondary sources" is presenting application of natural carbon isotopes to characterise carbonaceous aerosol composition in Beijing and Xi'an (two Chinese mega-cities). The title of the paper does not seem to fully reflect the subject of the article. In fact the spec-trum of issues discussed in the paper is much wider than "distinctions in primary versus secondary sources". While we are observing an increased interest in the topics related

to the study of air quality and its impact on the health of citizens, the development of various methods dedicated for identification the pollution sources is a very important topic that fits into the scope of the journal. Carbonaceous aerosols constitute an important fraction of air pollution observed in mega-cities. Application of carbon isotopes allows to identify the share and its temporal and spatial dynamics of different emission sources. In this context the paper address relevant scientific problem and demonstrates a possible solution based on comprehensive use of isotopic tracers applied for both elemental carbon (EC) and organic carbon (OC) fractions of carbonaceous pollutants. The presented methodology was applied for the short term measurement campaigns performed in two Chinese mega-cities having different emission sources structure. Authors demonstrated usefulness of this methodology for identifying share of the pollutants having different origin and presented an interesting data increasing the understanding of the differences between at first glance similar urban environments. The paper is well structured, the description of methodology is clear and in my opinion complete. Abstract contains a clear message of the paper. The results are correctly presented and in most parts well discussed. The quality and number of figures is correct. Presented results are discussed in relation to other studies appropriate referenced in the text. I recommend to publish the paper after a minor revisions.

Specific comments

The authors discuss in the paper a wide spectrum of issues including different contribution of coal, liquid fossil fuel and biomass combustion to elemental and organic fraction of carbonaceous aerosols, temporal (day-time vs. night time and haze events vs. clean periods) variability as well as spatial (location specific differences between Beijing and Xi'an cities) variability. In addition an issues related to primary and secondary organic aerosols are discussed. In such wide range of discussed aspects it is difficult to keep the description clear. Maybe a short summary following each part of the discussion would help reader to keep on track of the analysis.

The Methodology section is very detailed and well referenced but I found no information concerning possible mineral contamination (carbonates) of the collected samples is present. Were there any corrections to the mineral fraction contamination in the samples applied?

More detailed description of study sites (location, topography, typical emissions) would help to understand the differences in presented results between two cities.

Did the authors considered the admixture of bio-fuels into liquid fossil component in the context of F14Cliq.fossil parameter? Is it a case of Chinese liquid fuels market?

I don't see any clear reason for introducing equations 14 to 16.

---

## Referee Comment (RC2) · Anonymous Referee #1 · 3 Sep 2020

The measurement report by Ni et al. is an excellent manifestation of the measurement report paper. Despite not presenting strikingly new results, comparative studies involving lesser studied cities or regions are encouraged as long as the data analysis is done robustly and extensively. I particularly commend Bayesian MCMC simulation in addition to 14C analysis which enhances the results as well as excellent and informative Figures. The paper is very well written and can be published after addressing few minor comments.

Comments

[Figure]

Interactive
comment

Line 22. The term "clean" can hardly be attributed to mega-city environment. Perhaps "moderately polluted" or at least "relatively clean". Haze is typically related to poor visibility, so perhaps the use of "clear weather", which can be quite polluted too, can be more informative. The term "clean" is used throughout the text and I encourage finding a proper substitute.

Line 33. Field measurements.

Line 35. I think it should be stressed that better understanding of sources outside capital Beijing is needed for comparison as well as for more comprehensive understanding. Beijing was fairly well studied already.

Line 85. Please indicate similarity quantitatively as there is a proportionality issue commented later.

Line 149. Concentrations neither <100 nor <20 can be considered clean, especially that the two numbers differ by five times. If haze concentrations in Xian were defined >250, that is only 2-3 times different to clean, so certainly qualifies for moderate pollution. Furthermore, if Chinese national pollution standard is at 75ug/m3, concentration <100 can in no way qualify for clean.

Line 177. will be much higher.

Line 250. ...emissions are very low in winter.

Line 277. Why would large secondary formation from fossil sources be particularly favoured in Beijing only? With no plausible explanation it should be dismissed. Stronger contribution from coal combustion (both primary and secondary) sounds convincing given Beijing geographical location.

Line 287. ...by significantly larger...

Line 294. ...wide range of EC concentrations.

Line 301. same as above comment

---

## Author Comment (AC1) · 2 Oct 2020

We thank the reviewers for their thoughtful and valuable comments, which are very helpful for revising and improving our manuscript. The attached pdf file contains a detailed response to the points raised by the reviewers. The marked-up version of the manuscript is also included in the attached pdf file.

Please also note the supplement to this comment:
https://acp.copernicus.org/preprints/acp-2020-455/acp-2020-455-AC1-supplement.pdf

---

## Author Response (AR1)

We thank the reviewers for the helpful comments and providing us the opportunity to strengthen our research. We try to address all of them carefully. Below are point-to-point responses.

**Anonymous reviewer #1:**

The measurement report by Ni et al. is an excellent manifestation of the measurement report paper. Despite not presenting strikingly new results, comparative studies involving lesser studied cities or regions are encouraged as long as the data analysis is done robustly and extensively. I particularly commend Bayesian MCMC simulation in addition to $^{14}C$ analysis which enhances the results as well as excellent and informative Figures. The paper is very well written and can be published after addressing few minor comments.

**1)** Line 22. The term "clean" can hardly be attributed to mega-city environment. Perhaps "moderately polluted" or at least "relatively clean". Haze is typically related to poor visibility, so perhaps the use of "clear weather", which can be quite polluted too, can be more informative. The term "clean" is used throughout the text and I encourage finding a proper substitute.

**Response:** We agree with the reviewer that the "clean" periods we defined in this manuscript cannot fully represent the pollution levels, especially for "clean" periods in Xi'an. In Xi'an, as stated in Sect. 3.1 (page 6, line 172–173), even during clean periods we defined here, daily $PM_{2.5}$ concentrations (<100 µg m$^{-3}$) were higher than the Chinese pollution standard of 75 µg m$^{-3}$. However, in this study, we intent to use "clean" and "haze" to represent low and high pollution events, respectively, classified by $PM_{2.5}$ concentrations. With this study scope, we would prefer not to use the term "clear weather", because it is a weather condition, which cannot be identified by $PM_{2.5}$ concentrations.

In this study, we define clean days as daily average $PM_{2.5}$ concentrations < median concentrations in the winter heating season (15 November 2016 to 15 March 2017). If possible, we think it probably acceptable to keep the term "clean" to facilitate the writing, and to avoid the inconsistence between the discussion paper and the final one (e.g., "clean" are included in nearly all figures and tables as legend, text annotation etc.). In the revised manuscript, we explain quantitively the definition of "clean" in this study (changes are underlined):

(i) Abstract lines 16–19 (page 1)

"Here we combined the analysis of radiocarbon and the stable isotope $^{13}C$ to investigate the sources and formation of carbonaceous aerosols collected in two Chinese megacities (Beijing and Xi'an) during severe haze events of "red alarm" level from December 2016 to January 2017. The haze periods with daily $PM_{2.5}$ concentrations as high as ~400 µg m$^{-3}$ were compared to subsequent clean periods (i.e., $PM_{2.5}$ < median concentrations during the winter 2016/2017), with $PM_{2.5}$ concentrations below 100 µg m$^{-3}$ in Xi'an and below 20 µg m$^{-3}$ in Beijing."

(ii) Manuscript lines 101–103 (page 4)

Six samples _from haze and clean days_ were selected per sampling site for carbon isotope analysis (Tables S1 and S2). _We define clean days at each site as PM$_{2.5}$ < median concentration in the winter heating season from 15 November 2016 to 15 March 2017._

(iii) Figure caption (page 18)

**Figure 1.** Mass concentrations of OC and EC from fossil and non-fossil sources (OC$_{fossil}$, OC$_{nf}$, EC$_{fossil}$ and EC$_{bb}$) as well as fraction of non-fossil carbon in OC and EC ($f_{nf}$(OC) and $f_{bb}$(EC), respectively) for daytime and nighttime PM$_{2.5}$ samples in Xi'an, and 24h-integrated PM$_{2.5}$ samples in Beijing _during haze and clean periods during the measurement periods (2 December 2016 to 10 January 2017). For each city "haze" and "clean" are used to represent high and low pollution events, and clean days at each site are defined as days with PM$_{2.5}$ < median concentration in the winter heating season from 15 November 2016 to 15 March 2017._ Uncertainties of $^{14}$C-apportioned $f_{nf}$(OC) and $f_{bb}$(EC) are indicated but are too small to be visible. The data are shown in Table S3.

**2)** Line 33. Field measurements.

**Response:** Thank you for spotting out the typo. Corrected (page 2, line 36).

"_Field_ measurements show that carbonaceous aerosol contributes a significant fraction of PM$_{2.5}$ loading during severe haze events in China"

**3)** Line 35. I think it should be stressed that better understanding of sources outside capital Beijing is needed for comparison as well as for more comprehensive understanding. Beijing was fairly well studied already.

**Response:** We improved the introduction according to the comments and suggestions by the reviewer. The following underlined sentences are added in the introduction (page 2, line 39–41):

"Severe haze pollution with high PM$_{2.5}$ (i.e., particulate matter with aerodynamic diameter ≤ 2.5 µm) concentrations and reduced visibility occurs frequently during winter in China (An et al., 2019). Field measurements show that carbonaceous aerosol contributes a significant fraction of PM$_{2.5}$ loading during severe haze events in China (Huang et al., 2014; Elser et al., 2016; Liu et al., 2016). Therefore, a better understanding of the sources and atmospheric processes of carbonaceous aerosols is needed for mitigating haze pollution. _Many previous studies focus solely on Beijing, the capital of China. However, studies on other megacities are also needed for comparison as well as for a more comprehensive understanding of haze pollution in China._"

**4)** Line 85. Please indicate similarity quantitatively as there is a proportionality issue commented later.

**Response:** The revised text shows:

"Each composite sample consists of 2 12h (for Xi'an) or 24 h (for Beijing) filter pieces with similar $PM_{2.5}$ loading that agree within 20 % (Fig. S1)." (page 4, line 105–106)

**5)** Line 149. Concentrations neither <100 nor <20 can be considered clean, especially that the two numbers differ by five times. If haze concentrations in Xian were defined >250, that is only 2-3 times different to clean, so certainly qualifies for moderate pollution. Furthermore, if Chinese national pollution standard is at 75ug/m3, concentration <100 can in no way qualify for clean.

**Response:** Thank you for pointing this out. In this study, the haze periods with daily $PM_{2.5}$ concentrations as high as ~400 µg m$^{-3}$ were compared to subsequent clean periods, with $PM_{2.5}$ concentrations below 100 µg m$^{-3}$ in Xi'an and below 20 µg m$^{-3}$ in Beijing. We define clean days as daily average $PM_{2.5}$ concentrations < median concentrations in the winter heating season (15 November 2016 to 15 March 2017). This has been clarified in the revised manuscript, as addressed in the response to question 1.

We agree with the reviewer that in Xi'an $PM_{2.5}$ concentrations< 100 µg m$^{-3}$ quantifies "moderate pollution", as state in the Sect. 3.1: "In Xi'an, even during clean periods we defined here, daily $PM_{2.5}$ concentrations were higher than the Chinese pollution standard of 75 µg m$^{-3}$, reflecting severe air quality problems." (page 6, line 172–173). However, $PM_{2.5}$ concentration < 20 µg m$^{-3}$ in Beijing can be considered as clean, because it is smaller than the 24 h mean of 25 µg m$^{-3}$ suggested by the Air Quality Guidelines of World Health Organization (WHO, 2006).

In the revised manuscript, to avoid any confusing, the term "clean" is defined quantitively as daily average $PM_{2.5}$ concentrations < median concentrations in the winter heating season (15 November 2016 to 15 March 2017), and is explained and clarified several times, for example, in the abstract and in the main text when "clean" is used (see changes in response to question 1).

**6)** Line 177. will be much higher.

**Response:** Done (page 7, line 199).

"So even though biomass burning contributes a small portion of EC, its contribution to primary OC will be much higher."

**7)** Line 250. ...emissions are very low in winter.

**Response:** Done (page 9, line 279).

"The $OC_{o,nf}$ is not likely attributed to biogenic OC, because the biogenic emissions are very low in winter."

**8)** Line 277. Why would large secondary formation from fossil sources be particularly favoured in Beijing only? With no plausible explanation it should be dismissed. Stronger contribution from coal combustion (both primary and secondary) sounds convincing given Beijing geographical location.

**Response:** In this study, large secondary formation from fossil sources in Beijing is justified by larger $f_{\text{fossil}}$(SOC) in Beijing than in Xi'an (Fig. 6), where $f_{\text{fossil}}$(SOC) is the fraction of fossil SOC in total SOC of both fossil and non-fossil origins. However, the explanations for large secondary formation from fossil sources in Beijing are not clear yet, as the reviewer pointed out. To clarify, the following underlined sentences are added in the revised manuscript:

> "Compared to Xi'an, Beijing had significantly smaller differences between $f_{\text{bb}}$(EC) and $f_{\text{nf}}$(OC) (Fig. 1), which was also observed in previous studies during the haze event in January 2013 (Zhang et al., 2015). Comparing Eq. (14) with Eq. (16), this suggests either strong contribution from coal combustion in Beijing or large secondary formation from fossil sources, or both. The stronger contribution of coal combustion to OC in Beijing than in Xi'an was a direct consequence of a larger proportion of coal combustion in EC in Beijing, as demonstrated by the Bayesian MCMC results of EC (Sect. 3.2). The latter was further validated by the variation of SOC. The $f_{\text{fossil}}$(SOC) in Beijing was higher than that in Xi'an, despite the variations between haze and clean periods (Sect. 3.3). By combining $^{14}$C measurements with other state-of-art analytical techniques (e.g., aerosol mass spectrometry), Huang et al. (2014) also found that fossil OC was mostly secondary in nature in Beijing, and non-fossil SOC formation was dominant in Xi'an during a wintertime haze episode (i.e., Beijing had a larger $f_{\text{fossil}}$(SOC) than Xi'an). However, atmospheric mechanisms responsible for the enhancement in fossil-derived SOA formation in Beijing remain unclear." (page 11, line 316–319)

**9)** Line 287. ...by significantly larger...

**Response:** We have corrected the text as suggested:

> "This is validated by significantly larger $f_{\text{fossil}}$(SOC) during haze periods (76%−81%) than during clean periods (~55%; Sect. 3.3)." (page 11, line 326).

**10)** Line 294. ...wide range of EC concentrations.

**Response:** Corrected.

> "In Xi'an, $f_{\text{bb}}$(EC) was fairly constant during haze and clean periods, despite the wide range of EC concentrations. " (page 12, line 336)

**11)** Line 301. same as above comment (i.e., question 8)

**Response:** Due to the unclear mechanisms responsible for the enhancement in fossil SOC in Beijing (as addressed in response to question 8), we would like to avoid to highlight this point in the conclusion, thus delete the following underlined sentence in the conclusion of the revised manuscript:

"The differences between non-fossil fraction in OC and EC were smaller in Beijing and larger in Xi'an. " (page 12, line 343–344)

**References:**

WHO: Air Quality Guidelines: Global Update 2005: Particulate Matter, Ozone, Nitrogen Dioxide and Sulfur Dioxide, World Health Organization, 2006.

**Anonymous reviewer #2:**

The article by Ni et al entitled "Measurement report: Dual-carbon isotopic characterization of carbonaceous aerosol in Beijing and Xi'an: distinctions in primary versus secondary sources" is presenting application of natural carbon isotopes to characterize carbonaceous aerosol composition in Beijing and Xi'an (two Chinese mega-cities). The title of the paper does not seem to fully reflect the subject of the article. In fact, the spectrum of issues discussed in the paper is much wider than "distinctions in primary versus secondary sources". While we are observing an increased interest in the topics related to the study of air quality and its impact on the health of citizens, the development of various methods dedicated for identification the pollution sources is a very important topic that fits into the scope of the journal. Carbonaceous aerosols constitute an important fraction of air pollution observed in mega-cities. Application of carbon isotopes allows to identify the share and its temporal and spatial dynamics of different emission sources. In this context the paper address relevant scientific problem and demonstrates a possible solution based on comprehensive use of isotopic tracers applied for both elemental carbon (EC) and organic carbon (OC) fractions of carbonaceous pollutants. The presented methodology was applied for the short-term measurement campaigns performed in two Chinese mega-cities having different emission sources structure. Authors demonstrated usefulness of this methodology for identifying share of the pollutants having different origin and presented an interesting data increasing the understanding of the differences between at first glance similar urban environments. The paper is well structured, the description of methodology is clear and, in my opinion, complete. Abstract contains a clear message of the paper. The results are correctly presented and, in most parts, well discussed. The quality and number of figures is correct. Presented results are discussed in relation to other studies appropriate referenced in the text. I recommend to publish the paper after a minor revision.

**Response:** Following the reviewer's concern about the title, we change the title to better reflect the content of the manuscript and to be more specific (changes are underlined):

> **"Dual-carbon isotopic characterization of carbonaceous aerosol reveal different primary and secondary sources in Beijing and Xi'an during severe haze events"**

We would like to avoid including "day-night difference" in the title, because it is only studied in Xi'an. Furthermore, the day-night difference in sources of carbonaceous aerosol was slight in this study, especially during haze periods in Xi'an.

**Specific comments**

1) The authors discuss in the paper a wide spectrum of issues including different contribution of coal, liquid fossil fuel and biomass combustion to elemental and organic fraction of carbonaceous aerosols, temporal (day-time vs. night time and haze events vs. clean periods) variability as well as spatial

(location specific differences between Beijing and Xi'an cities) variability. In addition, issues related to primary and secondary organic aerosols are discussed. In such wide range of discussed aspects, it is difficult to keep the description clear. Maybe a short summary following each part of the discussion would help reader to keep on track of the analysis.

**Response:** The revised manuscript adds a short summary at the end of each result section.

(i) Sect. 3.1 Fossil and non-fossil contributions to EC and OC

"Overall, our $^{14}$C data show that fossil sources contribute more strongly to EC and OC in Beijing than in Xi'an, which is consistent with previous observations. Both in Beijing and Xi'an the fossil vs. non-fossil contributions to EC and OC did not change drastically during haze and clean periods. In Xi'an, a slight day-night difference was observed during clean periods, but disappeared during haze periods, suggesting a large accumulation of particles." (page 7, line 215–218)

(ii) Sect. 3.2 Fossil EC apportioned by stable carbon isotopes: coal vs. liquid fossil fuel

"In summary, complementing $^{14}$C with $\delta^{13}$C allows for quantitative constraints on EC sources: EC was dominated by liquid fossil fuel combustion (i.e., vehicle emissions) in Xi'an and by coal burning in Beijing, especially during clean periods. In Xi'an, no strong day-night differences in EC sources were observed during haze and clean periods." (page 9, line 248–250)

(iii) Sect. 3.3 Primary and secondary OC

"Together these results reveal the differences in primary and secondary OC in two Chinese megacities. The contribution of SOC to total OC increased from clean to haze periods in Xi'an. In contrary, SOC/OC ratios increased from haze to clean periods in Beijing, mainly due to increased SOC from non-fossil sources. SOC was dominated by fossil sources in Beijing but by non-fossil sources in Xi'an, especially during haze periods. In Xi'an, the day-night difference was larger during clean periods with less SOC at night." (page 10, line 289–293)

(iv) Sect 3.4 Difference between the fractions of non-fossil carbon in OC and EC

"In conclusion, this section discusses the factors that governs the differences between $f_{bb}$(EC) and $f_{nf}$(OC), and concludes that smaller differences suggest stronger contribution from coal combustion and/or larger secondary formation from fossil sources. This is further examined and validated by source apportionment results of EC and OC (Sect. 3.1–3.3)." (page 11, line 328–330)

**2)** The Methodology section is very detailed and well referenced but I found no information concerning possible mineral contamination (carbonates) of the collected samples is present. Were there any corrections to the mineral fraction contamination in the samples applied?

**Response:** Thank you for pointing this out. For fine particle PM$_{2.5}$, the interference of carbonate carbon with the OC and EC is negligible in most ambient (Chow and Watson, 2002; Querol et al., 2004), because carbonate is mostly in the coarse mode.

In this study, we did preliminary studies to check the possible interference of carbonate carbon. PM$_{2.5}$ samples with the highest calcium concentrations (associate with carbonate in most cases; Chow and Watson, 2002) in Xi'an, China were used to check the presence of carbonate carbon. Carbonate carbon in an aerosol sample can be verified if there is significant difference in TC (i.e., total carbon) mass before and after acidification (i.e., expose the sample to HCl vapor; NIOSH, 1999). Negligible carbonate carbon was found, because TC mass before and after acidification agrees each other within the measurement uncertainty. PM$_{2.5}$ samples in Beijing are less likely affected by carbonates, owing to lower dust concentrations (i.e., a major source of carbonate) than in Xi'an (Huang et al., 2014). In this study, we therefore did not correct the mineral contamination (carbonates), because the contamination is very small compared to the relatively larger OC and EC amounts for both mass determination and carbon isotopic analysis.

We add the above discussion in the Supplement S1. In the main text, we add (page 4, line 97–99):

> "Acidification to remove potential interferences from carbonates is not necessary, because carbonate carbon in PM$_{2.5}$ samples is found to be negligible, compared to the relatively larger OC and EC amounts for both mass determination and carbon isotopic analysis (Supplement S1)."

**3)** More detailed description of study sites (location, topography, typical emissions) would help to understand the differences in presented results between two cities.

**Response:** In the revised manuscript, we add the site descriptions in the method section (page 3, line 76–88):

> "Xi'an is the largest city in northwestern China, with over 8.8 million residents and 2.5 million vehicles in 2016 (Xi'an Municipal Bureau of Statistics and NBS Survey Office in Xi'an, 2017). Surrounded by Qinling Mountains to the south and the Loess Plateau to the north, days with low wind speed occur frequently in Xi'an, promoting the accumulation of air pollutants. Xi'an is now facing increased serious air quality issues due to the rapid increase of motor vehicles and energy consumption in the past two decades. Besides residential coal combustion, biomass burning is also a major emission source in Xi'an and its surrounding areas (i.e., Guanzhong Plain) for heating and cooking especially in winter (Zhang et al., 2014; Xu et al.,

2016). Beijing, the capital of China, is a megacity with over 21 million residents and 5.7 million vehicles in 2016 (Beijing Municipal Bureau of Statistics and NBS Survey Office in Beijing, 2017). Beijing is located in the Beijing-Tianjin-Hebei region, the most economically developed region in North China. However, the rapid economic growth and urbanization associated with heavy coal consumption and rapid increase usage of vehicles lead to the poor air quality in Beijing. Besides local emissions, regional transport of pollutants between neighboring cities also contributes to air pollution in Beijing (Zheng et al., 2015; An et al., 2019)."

The new citations are included in the revised reference list.

**4)** Did the authors consider the admixture of bio-fuels into liquid fossil component in the context of $F^{14}C_{liq.fossil}$ parameter? Is it a case of Chinese liquid fuels market?

**Response:** $F^{14}C_{liq.fossil}$ denotes $F^{14}C$ of liquid fossil fuel, used in Eq. (13) for source apportionment of EC (page 6). In China, $F^{14}C_{liq.fossil}$ has been approximated as zero in EC source apportionment studies (e.g., Andersson et al., 2015; Fang et al., 2018), because liquid fossil fuel combustion emits predominantly fossil carbon. However, in recent years a small fraction of bio-fuels has been added to liquid fossil fuel (e.g., gasoline and diesel) for on-road transportation. Nowadays, most of the biofuels are produced by USA (mainly corn ethanol) and Brazil (mainly sugarcane ethanol). The admixture of bio-fuels into liquid fossil fuel increases $F^{14}C_{liq.fossil}$, although it is still unclear to what extent it affects $F^{14}C_{liq.fossil}$.

Back to the winter 2016/2017 (the measurement period in this study), Beijing and Xi'an were not in the 11 provinces that adopted the fuel E10 gasoline (i.e., 10% biomass-based ethanol and 90% gasoline), the major bio-fuel for transportation in China (Hao et al., 2018). The development of bio-fuels in the transport sector in China has been restrained by food security concern and land availability (Zhang and Chen, 2015), because the bio-fuel in China is mainly derived from corn. EC from bio-fuels is a very minor fraction of EC in China (Zheng et al., 2019), hence it is not be considered in this study. This might change in the future if bio-fuels are used more extensively.

**5)** I don't see any clear reason for introducing equations 14 to 16.

**Response:** Equations 14 to 16 are introduced for illustrating what governs the differences between $f_{bb}(EC)$ and $f_{nf}(OC)$. Comparing Eq. (14) with Eq. (16), we tried to explain that (a) the differences between $f_{bb}(EC)$ and $f_{nf}(OC)$ depend on the differences between $\frac{(1+SOC_{fossil}/POC_{fossil})}{(1+OC_{o,nf}/POC_{bb})} \times \frac{r_{fossil}}{r_{bb}}$ and 1 (page 10, line 303), and (b) further elaborate on the statements that "With smaller $r_{fossil}$ than $r_{bb}$, similar $f_{nf}(OC)$ and $f_{bb}(EC)$ can result from larger secondary formation from fossil sources than non-fossil sources (i.e., $\frac{(1+SOC_{fossil}/POC_{fossil})}{(1+OC_{o,nf}/POC_{bb})} > 1$)" (page 11, line 305–306) and "Therefore, in a city where

biomass burning and coal combustion are the dominant pollution sources, $f_{nf}(OC)$ and $f_{bb}(EC)$ will be more similar than in a city where the main sources are biomass burning and vehicle emissions." (page 11, line 307–309). The underlined statements are not straightforward without introducing equations 14 to 16. Those equations are given, because they are not so common. Therefore, if the reviewer allows, we prefer to keep those equations.

---

## Author Response (AR2)

**Comments to the Author:**

The authors have reasonably well addressed the comments of the two anonymous referees and they have modified their manuscript accordingly. However, several alterations and corrections are needed for the Main text and Supplement before the manuscript can be published in ACP.

**Response:** Thank you for providing us the opportunity to revise and improve our manuscript. The comments on the main text and supplement are addressed accordingly. Detailed responses to the comments are provided in blue. Attached please also find the marked-up manuscript to track the changes in the revised manuscript.

**1. Comments on the main text and supplement**

**Main text:**

Line 496: Replace "201613401" by "114, E1054-E1061, https://10.1073/pnas.16130114".

**Response:** We add the article number, page ranges and doi number:

"Winiger, P., Andersson, A., Eckhardt, S., Stohl, A., Semiletov, I. P., Dudarev, O. V., Charkin, A., Shakhova, N., Klimont, Z., and Heyes, C.: Siberian Arctic black carbon sources constrained by model and observation, P. Natl. Acad. Sci. USA, 114, E1054-E1061, https://doi.org/10.1073/pnas.1613401114, 2017."

Lines 555-557: Several of the abbreviations in the abscissa of the Figure are only defined in Table S1. Therefore, the Figure caption should be extended and reference should be made in it to Table S1.

**Response:** Thank you for this point. The revised Figure caption shows (page 24):

"**Figure 6.** Fraction of fossil carbon in EC, OC and SOC ($f_{fossil}$(EC), $f_{fossil}$(OC) and $f_{fossil}$(SOC), respectively) during haze and clean periods in Xi'an and Beijing. Interquartile ranges (25th–75th percentile) of the median $f_{fossil}$(SOC) are shown as vertical bars in purple. Uncertainties of $^{14}$C-apportioned $f_{fossil}$(EC) and $f_{fossil}$(OC) are indicated but are too small to be visible. The studied samples include day and night samples in Xi'an (X) and 24 h integrated samples in Beijing (B) during haze periods ("H" samples) and clean periods ("C" samples). For details of each sample (*x*-axis), see Fig. 1 and Table S1."

We also make reference to Fig. 1, because Fig.1 has the same *x*-axis (i.e., sample name) as Fig.6. In Fig.1, the definition of each sample name is indicated in the plot area of the Figure.

**Supplement:**

Page S2, line 5 from bottom: Abbreviations and acronyms, here "PDFs", should be defined (written full-out) when first used. "PDFs" is only defined on page S3.

**Response:** In the revised Supplement, we present both the full name and abbreviation of "PDFs" when first used (page S2). After the abbreviation is defined, only "PDFs" is used (e.g., page S3).

**2. Response:** Alterations and corrections are made following the editor's comments. Below are the alterations made in the main text and supplement.

**For the Main text:**

Line 2: Replace "reveal" by "reveals".

Line 49: Replace "quantification" by "quantifying".

Line 52: Replace "provide" by "provides".

Line 93: Replace "IMPROVE_A" by "The IMPROVE_A".

Line 94: Replace "measurements" by "measurement".

Line 116: Replace "is smaller" by "are smaller".

Line 158: Replace "equal to" by "are equal to".

Line 163: Abbreviations and acronyms, here "PDFs", should be defined (written full-out) when first used. Since "PDFs" seems to stand for "probability density functions" and is not used elsewhere in the Main text, it should simply be replaced here by "probability density functions".

Line 177: Replace "is resulted from" by "results from the".

Line 210: Replace "vary spatially" by "varies spatially".

Line 230: Replace "sources," by "sources;".

Line 254: Replace "e.g.," by "i.e.,".

Line 265: Replace ", in contrast to" by "; in contrast".

Line 269: Replace ", mainly" by "; this mainly".

Line 318: Abbreviations and acronyms, here "SOA", should be defined (written full-out) when first used. Since "SOA" stands for "secondary organic aerosol" and is not used elsewhere in the Main text, it should simply be replaced here by "secondary organic aerosol".

Line 327: Replace "governs the" by "govern the".

Line 339: Replace "dominate fossil" by "dominating fossil".

**For the Supplement:**

Page S1, line 3: Replace "reveal" by "reveals".

Page S2, line 2: Replace "associate" by "associated".

Page S2, line 6: Replace "agrees each" by "agree with each".

Page S15, line 3: The abbreviated journal name should be in regular font instead of in italic.

**3. Besides adjustments made above, we checked the main text and Supplement, and found that several changes are needed:**

**Main text:**

**(a). Page 3, line 90 (the revised version).** The end of sampling date in Beijing should be 8 January 2017 (it was a written mistake as 7 January 2017 in the original manuscript), as shown in Fig. S1 and Table S1.

"In Beijing, the 24 h integrated (10:00 a.m. to 10:00 a.m. the following day) $PM_{2.5}$ was collected from 2 December 2016 to **8** January 2017."

**(b). Page 7, line 208.** The standard deviation (SD) of $f_{fossil}$(OC) in Xi'an should be 2%, not 3% in the original manuscript. The SD can be calculated from $f_{fossil}$(OC) values in Xi'an in Table S3. This correction will not affect any conclusion from this study. Sorry that we were not aware this typo before.

"The presented overall average $f_{fossil}$(OC) for winter 2016/2017 in Beijing ($66 \pm 3\%$) was higher than that in Xi'an ($46 \pm \mathbf{2}\%$)"

**Supplement:**

**(c). Title of Table S4 (page S12).** We add the abbreviation (POC$_{fossil}$) for primary OC from fossil sources in the title:

"**Table S4.** Concentrations ($\mu g\ m^{-3}$) of primary OC from biomass burning (POC$_{bb}$), primary OC from fossil sources (POC$_{fossil}$)...."

**(d). Title of Table S5 (page S13).** It should be "in different cities", not "in different seasons"

[revised manuscript text omitted]

PM$_{2.5}$ samples with the highest concentrations of calcium (associated with carbonate in most cases; Chow and Watson, 2002) in Xi'an, China were used to check the presence of carbonate carbon. Carbonate carbon in an aerosol sample can be verified if there is significant difference in TC mass before and after acidification (i.e., expose the sample to HCl vapor; NIOSH, 1999). Negligible carbonate carbon was found, because TC mass before and after acidification agrees with each other within the measurement uncertainty. PM$_{2.5}$ samples in Beijing are less likely affected by carbonates, owing to lower dust concentrations (i.e., a major source of carbonate) than in Xi'an (Huang et al., 2014). Therefore, acidification to remove 
[revised manuscript text omitted]

Chow, J. C. and Watson, J. G.: $PM_{2.5}$ carbonate concentrations at regionally representative Interagency Monitoring of Protected Visual Environments sites, J. Geophys. Res., 107( D21), 8344, doi:10.1029/2001JD000574, 2002.

Dusek, U., Monaco, M., Prokopiou, M., Gongriep, F., Hitzenberger, R., Meijer, H. A. J., and Röckmann, T.: Evaluation of a two-step thermal method for separating organic and elemental carbon for radiocarbon analysis, Atmos. Meas. Tech., 7, 1943–1955, https://doi.org/10.5194/amt-7-1943-2014, 2014.

Huang, R. J., Zhang, Y., Bozzeti, C., Ho, K. F., Cao, J. J., Han, Y., Daellenbach, K. R., Slowik, J.G., Platt, S. M., Canonaco, F., Zotter, P., Wolf, R., Pieber, S. M., Bruns, E. A., Crippa, M., Ciarelli, G., Piazzalunga, A., Schwikowski, M., Abbaszade, G., SchnelleKreis, J., Zimmermann, R., An, Z., Szidat, S., Baltensperger, U., El Haddad, I., and Prévôt, A. S. H.: High secondary aerosol contribution to particulate pollution during haze events in China, Nature, 514, 218–222, https://doi.org/10.1038/nature13774, 2014.

Ni, H., Huang, R.-J., Cao, J., Liu, W., Zhang, T., Wang, M., Meijer, H. A. J., and Dusek, U.: Source apportionment of carbonaceous aerosols in Xi'an, China: insights from a full year of measurements of radiocarbon and the stable isotope $^{13}C$, Atmos. Chem. Phys., 18, 16363–16383, https://doi.org/10.5194/acp-18-16363-2018, 2018.

NIOSH: Method 5040 Issue 3 (Interim): Elemental carbon (diesel exhaust), in: NIOSH Manual of Analytical Methods; National Institute of Occupational Safety and Health: Cincinnati, OH, 1999.

Zenker, K., Vonwiller, M., Szidat, S., Calzolai, G., Giannoni, M., Bernardoni, V., Jedynska, A. D., Henzing, B., Meijer, H. A., and Dusek, U.: Evaluation and Inter-comparison of oxygen-based OC-EC separation methods for radiocarbon analysis of ambient aerosol particle samples, Atmosphere, 8, 226, https://doi.org/10.3390/atmos8110226, 2017.

Zhang, Y. L., Li, J., Zhang, G., Zotter, P., Huang, R.-J., Tang, J.-H., Wacker, L., Prévôt, A. S. H., and Szidat, S.: Radiocarbon-based source apportionment of carbonaceous aerosols at a regional background site on Hainan Island, South China, Environ. Sci. Technol., 48, 2651–2659, https://doi.org/10.1021/es4050852, 2014.

Zhang, Y. L., Huang, R. J., El Haddad, I., Ho, K. F., Cao, J. J., Han, Y., Zotter, P., Bozzetti, C., Daellenbach, K. R., Canonaco, F., Slowik, J. G., Salazar, G., Schwikowski, M., Schnelle-Kreis, J., Abbaszade, G., Zimmermann, R., Baltensperger, U., Prévôt, A. S. H., and Szidat, S.: Fossil vs. non-fossil sources of fine carbonaceous aerosols in four Chinese cities during the extreme winter haze episode of 2013, Atmos. Chem. Phys., 15, 1299–1312, https://doi.org/10.5194/acp-15-1299-2015, 2015.